# Oral Microbiome in Relation to Periodontitis Severity and Systemic Inflammation

**DOI:** 10.3390/ijms22115876

**Published:** 2021-05-30

**Authors:** Adelina S. Plachokova, Sergio Andreu-Sánchez, Marlies P. Noz, Jingyuan Fu, Niels P. Riksen

**Affiliations:** 1Department of Dentistry, Radboud University Medical Center, 6525 EX Nijmegen, The Netherlands; 2Department of Genetics, University Medical Center Groningen, University of Groningen, 9700 RB Groningen, The Netherlands; s.andreu.sanchez@umcg.nl (S.A.-S.); j.fu@umcg.nl (J.F.); 3Department of Pediatrics, University Medical Center Groningen, University of Groningen, 9700 RB Groningen, The Netherlands; 4Department of Internal Medicine and Radboud Institute for Molecular Life Science (RIMLS), Radboud University Medical Center, 6525 GA Nijmegen, The Netherlands; marlies.noz@radboudumc.nl

**Keywords:** periodontitis, systemic inflammation, oral microbiome, cardiovascular disease

## Abstract

Systemic inflammation induced by periodontitis is suggested to be the link between periodontitis and cardiovascular disease. The aim of this work was to explore the oral microbiome in periodontitis in relation to disease severity and systemic inflammation. The saliva and subgingival microbiome from periodontal pocket samples of patients with severe (*n* = 12) and mild periodontitis (*n* = 13) were analyzed using metagenomic shotgun sequencing. The taxa and pathways abundances were quantified. The diversity was assessed and the abundances to phenotype associations were performed using ANCOM and linear regression. A panel of inflammatory markers was measured in blood and was associated with taxa abundance. The microbial diversity and species richness did not differ between severe and mild periodontitis in either saliva or periodontal pockets. However, there were significant differences in the microbial composition between severe and mild periodontitis in the subgingival microbiome (i.e., pocket samples) and, in a lower grade, in saliva, and this is positively associated with systemic inflammatory markers. The “red complex” and “cluster B” abundances in periodontal pockets were strongly associated with inflammatory markers interleukin-6 and the white blood cell count. Our data suggest that systemic inflammation in severe periodontitis may be driven by the oral microbiome and may support the indirect (inflammatory) mechanism for the association between periodontitis and cardiovascular disease.

## 1. Introduction

Clinical research has identified an association between periodontal disease (PD) and cardiovascular disease (CVD). The main suspected mechanisms driving this association are infectious (direct effect) and inflammatory (indirect effect) [1,2]. The indirect mechanism suggests that periodontal inflammation caused by oral bacteria induces the production of inflammatory mediators in the systemic circulation. To date, the association between the oral microbiome and systemic inflammation has not been extensively investigated. It is unknown whether oral bacteria that may cause systemic inflammation are part of the saliva, subgingival periodontal pocket communities, or both. Most periodontal studies have been performed only on the subgingival microbiome and have yielded contradictory results. For example, the subgingival bacterial burden of 11 species was not found to be associated with white blood cell counts or C-reactive protein (CRP) [3]. However, in another study [4], the phyla *Firmicutes* and *TM 7*, along with 18 other individual taxa, were positively associated with a systemic inflammatory score, whereas the established periodontal pathogens from the “red complex” (i.e., *Porphyromonas gingivalis*, *Tannerella forsythia*, and *Treponema denticola*) [5] were found not to be associated [4]. In addition, current evidence for the correlation between the oral microbiota and systemic inflammation comes indirectly from intervention studies that report a reduction in systemic inflammation after periodontal therapy [6]. However, this correlation may not be directly related to the oral microbiome.

To date, there are many studies that have focused on the oral microbiome in PD that use metagenomic technologies highlighted in two recent reviews [7,8]. However, most of these studies are based on 16S rRNA amplicon sequencing. While amplicon sequencing amplifies and sequences a small region of the ribosomal sequence, metagenomic shotgun-sequencing (MGS) enables the sequencing of entire genomes. Thus, MGS analyses provide more information about the microbial community investigated, such as predictions of the metabolic potential or a higher taxonomic resolution than 16S. To our knowledge, there are limited studies on the oral microbiome in PD using MGS [9,10], and none have investigated the possible association between the oral microbiome and systemic inflammation. 

In view of all the above mentioned issues, the aim of our study is to comprehensively explore the oral microbiome present in saliva and periodontal pockets (i.e., subgingival microbiome) in patients with clinically diagnosed PD using MGS, and its relationship with a measured panel of circulating inflammatory markers. In addition, we present a targeted analysis focusing on known periodontal pathogens and oral bacteria described to be present in human atherosclerotic plaques. Our data show distinguishable differences in the composition of the subgingival microbiome (i.e., pocket samples) between severe and mild PD. Such microbial differences are associated with systemic inflammatory markers, suggesting the role of those taxa in promoting systemic inflammation.

## 2. Results

### 2.1. Study Participants

The patient cohort involved 13 mild PD and 12 severe PD patients. According to the periodontal classification 2017 [11,12], there were 10 cases of Stage I PD and 3 cases of gingivitis in the group with mild PD (*n* = 13). The severe PD group (*n* = 12) consisted of 9 cases of Stage III PD, 2 cases of Stage IV PD, and 1 case of Stage II PD. The demographic characteristics of both groups are presented in Table 1. There were no significant differences in the CVD risk factors, namely BMI, hypertension, and smoking status, between the groups. There was a trend toward higher circulating concentrations of IL-6 and IL-1RA in the patients with severe PD compared to the mild PD (*p* < 0.1), whereas circulating WBC and hsCRP were similar between groups. In addition, periodontal inflammation, measured as the [^18^F]FDG uptake in the periodontium on PET/CT scans, tended to be higher in the group of severe PD (Table 1). 

### 2.2. Microbiome Signatures 

The microbial composition and diversity of each community (i.e., saliva and subgingival microbiome in periodontal pockets) were quantified on all taxonomic levels (i.e., from phylum to species). Differences between saliva and periodontal pocket (also referred to as pocket) communities were identified. Subsequently, the diversity and bacterial taxonomy were associated with disease severity (i.e., mild and severe PD). Bacterial metabolic pathway abundances in saliva and pockets were compared between mild and severe PD. Finally, bacteria and metabolic pathways associated with disease severity were studied for relationships with the markers of systemic inflammation.

#### 2.2.1. Composition of the Oral Microbiome and Its Association with PD Disease Severity

The microbial composition and functional profile were quantified in 37 samples, comprising both saliva and periodontal pockets, using a marker-based mapping approach. In addition, 15 samples belonged to saliva (8 mild PD/7 severe PD) and 22 were pooled subgingival samples from periodontal pockets (11 mild PD/11 severe PD) (Appendix A). A total of 11 phyla, 91 genera, 235 species, and 340 microbial metabolic pathways were detected in the saliva and pocket samples from all patients. The phylum level composition of saliva and pocket samples was comparable between participants with mild and severe PD (Figure 1A). Assessment of the bacterial composition of the most abundant species showed *Corynebacterium matruchotii* to be significantly more abundant in pocket samples of participants with mild PD compared to those with severe PD (Figure 1B). 

#### 2.2.2. Diversity of the Microbiome and Its Association with PD Disease Severity 

No significant difference in alpha-diversity (Kruskal–Wallis, *p* = 0.81), which measures the species diversity within one sample estimated with the Shannon index, or species richness (Kruskal–Wallis, *p* = 0.39) was found between mild and severe PD and tissue source (i.e., saliva or pockets) groups (Figure 2). Beta-diversity, an index for the species diversity between two samples, estimated using the Bray–Curtis dissimilarity index, showed separation between the saliva and pocket samples in a hierarchical cluster dendrogram and ordination plots (Figure 3A,B). In periodontal pockets, severe PD samples were shifted from the mild PD samples (PERMANOVA 2000 permutations, *p* = 0.05) (Figure 3A). After independently analyzing pocket and saliva samples, we observed the microbial composition in mild PD pockets to be significantly different compared to those in severe PD (PERMANOVA 2000 permutations, *p* = 0.03). In saliva samples, there was no significant difference in composition between mild and severe PD (PERMANOVA, *p* = 0.25) (Figure 3C).

We further analyzed the beta-diversity associations with different phenotype markers via PERMANOVA (Figure 3C), showing significant associations of bacteria in periodontal pockets with different markers of systemic inflammation.

#### 2.2.3. Individual Taxa and Metabolic Pathways Associated with Disease Severity 

Next, we used ANCOM to identify taxa associated with periodontal disease (Appendix A). In saliva samples, we identified 11 species present exclusively in patients with severe PD, yet in a low proportion of participants. Among them, the most abundant species were *Parvimonas micra*, *Anaeroglobus germinatus*, and *Bulleida extructa*, which have previously been associated with different health conditions (Table 2). 

In pocket samples, the abundances of *Actinomyces johnsonii*, *Corynebacterium matruchotii*, and *Capnocytophaga unclassified* were significantly lower in severe PD, whereas the abundances of *Porphyromonas endodontalis*, *Eubacterium brachy*, and *Eubacterium saphenum* were significantly higher (Appendix A).

Furthermore, we expanded our abovementioned species-level analyses by also investigating whether consistent bacterial strain differences exist between mild and severe PD. We obtained within-species genetic distance data for only 26 species where at least 3 samples were available. From those, the majority included less than 10 samples. PERMANOVA analysis did not show significant associations between phylogenetic distance and either severity or sample location. The only significant association, *Human_endogenous_retrovirus_K*, however, had only 1 mild-PD sample, which hampered a proper assessment of the existence of a consistent phenomenon (Methods) (Appendix A).

After comparing bacterial taxa differences, bacterial metabolic pathway abundances in saliva and pockets were compared between the two groups (Appendix A). In saliva samples, there were no significant differences, while in pocket samples, 6 pathways were found in significantly lower abundances in severe PD compared to mild PD. 5 of those were anabolic pathways, carrying out the production of glutamate, heme, or the amino acids isoleucine and valine. The final pathway was involved in the pyruvate fermentation. 

In addition, individual taxa and pathways were studied in relation to the probing pocket depths, which is a continuous variable that can be used as a proxy for disease severity. Similar associations as with PD severity were identified (Appendix A).

#### 2.2.4. Identification of Bacteria and Bacterial Clusters Previously Reported in the Literature to Be Associated with PD and CVD

We tested whether certain groups of periodontal pathogens, known as the pathogenic clusters driving microbial dysbiosis and inflammation in periodontitis, were more abundant in individuals with severe PD than in those with mild PD. Additionally, we identified and compared between PD groups a series of taxa that had been previously associated with clinical PD, dyslipidemia, and atherosclerotic plaques (Table 2).

The “red complex” (i.e., *Porphyromonas gingivalis*, *Tannerella forsythia*, and *Treponema denticola*) [5] was found significantly increased both in saliva (Wilcoxon test, *p* = 0.03) and in pocket samples (Wilcoxon test, *p* = 8 × 10^−3^) of patients with severe PD (Figure 4A).

“Cluster A,” reported to be representative for a mild periodontitis phenotype [23] that consists of the bacterial genera *Campylobacter*, *Corynebacterium*, *Fusobacterium*, *Leptotrichia*, *Tanerella* and *Saccaribacteria*, was not found to be significantly different between the defined PD severity levels in our study (Appendix A). “Cluster B,” reported to be higher with severe PD, was found significantly more abundant in the periodontal pockets of the severe PD group (Wilcoxon test, *p* = 0.02) (Appendix A). This cluster is represented by *Porphyromonas gingivalis*, *Tannerella forsythia*, and *Treponema denticola* (i.e., red complex), in addition to *Filifactior alocis*, *Treponema* spp., and *Fretibacterium* spp. [23].

Next, we focused on oral bacterial taxa that had been previously detected in human atherosclerotic plaques. The combined abundance of genera *Propionibacterium* and *Parvimonas* was significantly higher in saliva (Wilcoxon test, *p* = 0.02) and pocket samples (Wilcoxon test, *p* = 0.01) of the group with severe PD (Figure 4B). 

Additionally, we identified other lineages of interest in CVD to be different between patients with mild and severe PD. *Anaeroglobus geminatus*, the abundance in the oral cavity of which was found to be associated with symptomatic atherosclerosis [15], was detected only in the saliva of patients with severe PD. The same was observed for the key stone pathogen *Porphyromonas gingivalis*.

#### 2.2.5. PD Bacteria and Bacterial Metabolic Pathways Associated with Systemic Inflammation

We explored whether bacteria and metabolic pathways more abundant in severe PD were also associated with systemic inflammation (Appendix A). For this purpose, we correlated using linear models both species and pathways that differed between mild and severe PD in our study with the markers for systemic inflammation, and the uptake of glucose in the periodontium as a measurement of periodontal tissue inflammation by PET-CT scans (Appendix A). In saliva samples, the abundance of *Porphyromonas gingivalis* was significantly associated with the uptake of glucose in the periodontium (F-test *p* = 3 × 10^−3^), measured by PET-CT scans and expressed as SUVmean. 

In periodontal pockets, 7 significant associations were found. *Corynebacterium matruchotii* was negatively associated with circulating inflammation markers IL-6, IL-1Ra, and the leukocyte count (WBC). *Eubacterium saphenum* was positively correlated with WBC and IL-1Ra. In addition, *Eubacterium brachy* and *Porphyromonas endodontalis* showed trends for associations with IL-6, IL-1Ra, WBC, and the SUVmean of periodontium. 

Next, we checked how bacterial clusters that were increased in patients with severe PD were associated with the markers of systemic inflammation. The “red complex” was significantly associated with IL-6 (F-test *p* = 0.01), IL-1Ra (F-test *p* = 0.02), and WBC (F-test *p* = 0.03) in pocket samples, whereas, in saliva, no significant associations were found. “Cluster B” in pocket samples was similarly associated with WBC (F-test *p* = 6 × 10^−3^), IL-6 (F-test *p* = 5 × 10^−4^), and IL-1Ra (F-test *p* = 6 × 10^−3^). The cluster of *Propionibacterium propionicum*, *Parvimonas micra*, and *Parvimonas unclassified* was only significantly correlated with IL-1Ra in saliva (F-test *p* = 0.03).

Oral bacteria reported in the literature to be present in high abundance in human atherosclerotic plaques were not found to be associated with systemic inflammation in pockets, while they were significantly associated with IL-1Ra (pg/mL) (F-test *p* = 0.02) in saliva (Appendix A). Additionally, we found three pathways in pocket samples, which were negatively associated with severe PD that also showed negative associations with the inflammation markers, namely, the alanine biosynthesis pathway, heme biosynthesis pathway, and heme biosynthesis pathway from glutamate. All of them were negatively correlated with WBC, IL-6 (only a nonsignificant trend in the heme biosynthesis pathways) and IL-1Ra (Appendix A). 

## 3. Discussion

To the best of our knowledge, this is the first metagenomic analysis of the oral microbiome of saliva and periodontal pockets in patients with PD in relation to systemic inflammation. We found out that microbial diversity and species richness do not differ between severe PD and mild PD in either saliva or periodontal pockets. However, there were significant differences in the microbial composition between severe and mild PD in the subgingival microbiome and, in a lower grade, in saliva, and this was positively associated with systemic inflammatory markers.

We compared individuals with severe and mild PD, matched for known risk factors of CVD (e.g., hypercholesterolemia, hypertension, smoking, and BMI), and we measured circulating markers of inflammation. This allowed us to explore solely (i.e., eliminating confounders) the effect of the oral microbiota on systemic inflammation. Moreover, we studied the severity of PD defined according to the new classification of periodontal conditions 2017 [12]. Although the groups of mild and severe PD were not homogenous, our results provide insight into the microbiological profile and functionality of PD with different stages of severity.

We found PD severity to be associated with beta- and not with alpha-diversity in subgingival periodontal pocket samples. This is in accordance with another study that used MGS and compared subgingival samples of stable PD, progressive PD, and healthy patients [17]. Despite the different classification criteria for PD between both studies, and the lack of a healthy group in our investigation, it appears that alpha-diversity can be used as an indicator of microbial flux in a state of dysbiosis. As, in our study, both groups have microbial dysbiosis as being diagnosed with PD, the absence of significant alpha-diversity differences might be expected. Moreover, our findings support the current concept that PD is associated with ecological shifts in community structures rather than shifts in members of this microbial community [14]. Changes between disease status do not necessarily result from the replacement of health-associated species, but from the rise in new dominant species presented previously in low frequencies. This is in accordance with our observations; for instance, in periodontal pockets *Porphyromonas endodontalis*, *Eubacterium saphenum*, and *Eubacterium branchy* increased their abundance simultaneously with PD severity. These bacteria, together with *Filifactor alocis*, were also associated with the deepest probing pocket depths, which corroborates other studies [4,16,17,18,24] and supports their recognition as newly identified periodontal pathogens [16]. Moreover, *Porphyromonas endodontalis*, *Eubacterium brachy*, and *Eubacterium saphenum* were positively associated with the markers of systemic inflammation. This requires special attention and further investigation, especially because *Porphyromonas endodontalis* is known primarily to be associated with endodontal infections [25], which also frequently occur in patients without PD.

Interestingly, *Corynebacterium matruchotii* was negatively associated with severe PD, as well as with the inflammatory markers. This finding, together with the knowledge that *Corynebacterium matruchotii* is typical for healthy periodontal sites [16], suggests that it may be a bacterium beneficial for periodontal and systemic health, or a marker of a healthy microbial community. 

In saliva samples, the beta-diversity of the microbiome was not different between the two PD groups. The small sample size might explain why we did not detect significant differences. However, an interesting observation was that oral bacteria that were previously reported in human atherosclerotic plaques, such as *Porphyromonas gingivalis* and *Anaeroglobus germinates* [15], were found present only in saliva of patients with severe PD. For example, *Anaeroglobus* was detected in 3 out of 7 individuals with severe PD but not in the group with mild PD. Furthermore, *Porphyromonas gingivalis* was significantly associated with FDG-uptake in the periodontium, as measured by PET-CT scans, which once again, supports its etiological key role in PD. These observations require further investigation, especially how bacteria present in the saliva may enter the bloodstream and become encapsulated in the atherosclerotic plaques.

It is known that pathogenic species do not usually play a role as a single pathogenic entity [5], but that relationships among several species determine the pathogenic role of the microbiota. Our results further support this theory. Both bacteria of the Socransky’s “red complex” [5] and “cluster B” from Hong and co-workers [23] were found significantly enriched in the saliva and pockets of patients with severe PD compared to individuals with mild PD. Moreover, their abundance in subgingival plaque samples was also positively associated with circulating markers of systemic inflammation (e.g., IL-6 and WBC). Our findings suggest the important role of these bacteria not only for PD, but also in systemic inflammatory diseases. It has been proposed that periodontal keystone pathogens may play a role in the etiopathogenesis of diseases with chronic low-grade inflammation, such as cardiovascular diseases [26]. 

To further explore the association between PD and CVD, we studied clusters reported in the literature as isolated from atherosclerotic plaques. Genera *Propionibacterium* and *Parvimonas*, detected exclusively in atheroma plaques [13], were significantly increased in saliva and pockets in the group with severe PD. Interestingly, the combined abundance of *Propionibacterium propionicum*, *Parvimonas micra*, and *Parvimonas unclassified* in saliva was correlated with the marker IL-1Ra, which is an anti-inflammatory factor. Other oral bacteria, detected in human atherosclerotic plaques in symptomatic and asymptomatic CVD patients, such as *Porphyromonadaceae*, *Bacteroidaceae*, *Micrococcaceae*, *Streptococcaceae* [27], *Proteobacteria*, *Actinobacteria* [28], *Veilllonella*, and *Streptococcocus* [21] appeared not to be associated with systemic inflammation in our study. These bacteria belong to the commensal microbiota, and some of them are overabundant (*Proteobacteria* and *Actinobacteria)* in periodontal health [4]. 

Elaborating further on the role of bacteria for the pathogenesis of PD and associated systemic chronic inflammatory diseases, the “infectious burden” (i.e., total pathogen burden), comprising the aggregate number of pathogens infecting the individual, is considered to have a stronger association with CVD than the exposure to a single microbiome [21]. Therefore, all sources of bacteria (e.g., gut microbiome) should be considered when such an association is discussed [13,29]. The scope of the current study was only the oral microbiome. However, it could not be excluded that the gut microbiome of patients with severe PD is different to those without and, thus, could affect the systemic inflammation. 

MGS gave us the advantage to explore not only “who were present”, but also “what they did” [30]. Examining bacterial metabolic pathways can provide greater insight into functional differences between health (symbiotic) and a periodontitis (dysbiotic) profile. In our study, both groups had a dysbiotic profile due to the presence of disease, whereas the only difference was the severity of PD. This might explain the absence of big differences between groups. In pockets, 3 pathways were negatively associated with severe PD, namely, the alanine biosynthesis pathway, heme biosynthesis pathway, and heme biosynthesis pathway from glutamate. This is in agreement with reports that the disease-associated subgingival microbiome encodes more metabolic degradation processes than biosynthesis does [9,10]. Additionally, the underrepresented microbial metabolic pathways in severe PD showed negative associations with the inflammatory markers WBC and IL-6. It is known that periodontal pathogens are proteolytic, amino-acid degrading, and require hemin for their growth. The decrease in glutamate concentration in crevicular fluid is reported in PD [31]. Based on that and our findings, we may speculate that bacteria involved in pathways for the biosynthesis of amino-acids and heme (proteogenic and hemogenic) are beneficial for the absence of systemic inflammation. Other pathway differences related with PD might be masked by the absence of a healthy control group. 

Despite several limitations of our study, the small sample size, and the absence of healthy controls (i.e., participants with periodontal health), we have been able to pinpoint lineages and functional features involved in the pathophysiology of PD and associated with systemic inflammation. Our observations add data supporting the indirect (inflammatory) mechanism for the association between PD and CVD. 

## 4. Materials and Methods

### 4.1. Study Participants

In this study, 25 participants (40–80 years) were recruited among patients of the department of Dentistry, Radboud University Medical Center, The Netherlands. This study was an exploratory study, and hence, no sample size calculation had been performed. Exclusion criteria were previous CVD, auto-immune diseases (including diabetes mellitus), chronic immunomodulatory drug use, kidney disease (MDRD < 45 mL/min), or liver disease (ALAT > 135 U/L). In addition, patients were excluded with infection (>38.5 °C or antibiotic treatment), hospital admission, or vaccination within 1 month prior to study entry. The study protocol was approved by the Institutional Review Board Arnhem/Nijmegen, the Netherlands, with number 2017.3431/NL 61840.091.17. Approval was granted on 10 November 2017. All participants gave written informed consent. 

### 4.2. Clinical Periodontal Examination

The Dutch Periodontal Screening Index (DPSI) and probing pocket depths (PPD) were recorded by an experienced periodontist [AP]. Subjects with DPSI scores of 0, 1, and 2 were classified as having no PD (category A), those with a DPSI score of 3 as having mild PD (category B), and subjects with DPSI scores of 3+ and 4 as having severe PD (category C) [32,33]. The presence of alveolar bone loss on paired bitewings, defined as a > 2 mm distance between the cement–enamel junction and alveolar bone crest [34], was used to confirm the clinical diagnosis. Participants with DPSI category C and alveolar bone loss were assigned to the severe PD group. Subjects with DPSI category B with or without minor radiographic alveolar bone loss were categorized in the mild PD group. Participants of both groups were matched on shared risk factors for CVD such as age, sex, body mass index (BMI), and smoking. The case definitions for PD severity according to the new periodontal classification 2017 [11] were defined in both groups. In addition, the mean PPD and deepest PPD (mm) were calculated and defined based on all measured and recorded probing pockets depths.

### 4.3. Markers of Systemic Inflammation

Circulating cell counts in peripheral blood were measured by Sysmex analyser (Sysmex). Plasma high-sensitivity CRP (hsCRP) concentrations were obtained using an enzyme-linked immunosorbent assay (ELISA, R&D). Circulating interleukin (IL)-1Ra, and IL-6 concentrations were measured using a SimplePlex cartridge on the Ella platform (Protein Simple, San Jose, CA, USA). 

### 4.4. Periodontal Tissue Inflammation on [^18^F] FDG PET and Low-Dose CT Scanning

After adhering to a 24 h low-carbohydrate diet and 6 h of fasting, participants underwent 2’-deoxy-2’-[^18^F]fluoro-D-glucose positron emission tomography ([^18^F]FDG PET) (Appendix B) with low-dose noncontrast-enhanced CT on a dedicated Siemens Biograph 40 mCT scanner (Siemens Healthineers, Erlangen, Germany). Analyses were performed on reconstruction settings according to European guidelines [35] using Inveon Research Workspace v4.2 software. See Appendix A for details.

Periodontal tissue inflammation was determined by the [^18^F] FDG-uptake in the left and right periodontium of maxilla-to-mandible. The mean standardized uptake value (SUV_mean_) was calculated after correction for weight and [^18^F] FDG-dose (MBq).

### 4.5. Demographic and Biomarker Comparison

Mild and severe PD patients were compared for continuous variables using an independent samples T-test (if normally distributed according to Shapiro–Wilk test) and for categorical variables using a Χ^2^ test. For the circulating cytokines, chemokines, and periodontal inflammation, one-way ANCOVA was conducted to control for age differences between mild and severe PD. Beforehand, outliers were removed (if ±3 SD from Z-scores) and, thereafter, log(10)-transformed. Missing values were not imputed. A two-sided *p*-value < 0.05 was considered statistically significant.

### 4.6. Microbial Sample Collection 

Samples for shotgun sequencing were collected between 2 and 5 pm, and before any dental treatment had been performed. Participants were asked to refrain from coffee, cheese, and chlorhexidine usage and stay sober 1 h before sample collection. A modified protocol from Belstrøm et al. 2017 was used [36]. First, 2–5 mL stimulated saliva was collected after 30 s of chewing on a parafilm. Next, subgingival plaque samples were taken from the deepest PPD with bleeding using sterile paper points, as previously described [36]. We tried to choose standardized sites for collection of the subgingival microbiome. Samples were directly stored on ice in sterile Powerbead Solution (Qiagen, Hilden, Germany). Microbial analysis was performed on pooled subgingival samples (i.e., combining paper points from each site in the same tube). 

### 4.7. Metagenomic Shotgun Sequencing

Microbial DNA was extracted from saliva and pocket samples using DNeasy^®^ UltraClean^®^ Microbial Kit (Qiagen, Hilden, Germany), according to the manufacturer’s instructions. We performed automated isolation with Qiacube (Qiagen, Hilden, Germany), also according to the manufacturer’s protocol. The DNA concentration was determined using the Qubit DNA HS assay (Qiagen, Hilden, Germany), and the quality using Nanodrop technology. The DNA input for all samples was normalized, and libraries were sequenced in a single run. Library preparation was performed using Shotgun Metagenomic Sequencing with Illumina HiSeq sequencing by Novogene (Illumina, San Diego, CA, USA).

We extracted a mean of 19.7 million paired-end sequencing reads per sample (minimum of 15.1 × 10^6^ to maximum of 28.8 × 10^6^). FastQ files were preprocessed using KneadData v0.5.1 with the Homo_sapiens_Bowtie2_v0.1 database for removal of human reads (Appendix A) and default settings. Microbial composition profiling was performed with MetaPhlan2 v2.6.0 [37], and the functional profile of the microbial communities using HUMAnN2 v0.11.1 [38]. HUMAnN2 maps reads to a reference panel of gene families from the UniProt Reference Clusters [39] (UniRef90), which can be merged into pathways from the MetaCyc pathway database [40,41]. 

### 4.8. Data Analysis and Statistics

Data analysis was performed using R v3.6.2. We performed the following analysis on different taxonomic levels (species, genus, and phylum).

Within-sample diversity (alpha-diversity) was estimated as Shannon entropy (vegan v2.5-6 [42]). Differences in diversity between clinical groups were tested using Dunn’s test.

We calculated the Bray–Curtis dissimilarity [43] matrix from the taxonomy abundance using the vegan package. Principal components of the matrix were calculated using ape v5.3 [44]. We performed several PERMANOVA (vegan package) in the dissimilarity matrix (2000 permutations) to estimate the effect of disease status, PPDdeepest, PPDmean, IL-1Ra, IL-6, hsCRP, and WBC on the microbial community, while accounting for sample origin (saliva/pocket) and the number of reads available after QC. Hierarchical clustering of samples was performed based on the dissimilarity matrix.

Individual taxa differences (of taxa present in over 20% of samples in a minimal mean relative abundance of 0.01%) between the mild and the severe PD groups were examined independently in pocket and saliva samples using the ANCOM procedure [45], as implemented in [46]. Structural zeros (i.e., the absence of a taxon due to the underlying niche community) were assessed based on the total absence in one group but presence in the other. We used the function feature_table_pre_process (out_cut = 0.05, zero_cut = 0.90), for further removal of taxa considered as outliers or with low prevalence. Effect sizes are calculated from a linear model using the taxa’s centered log-ratio transformation (clr) [47] relative abundance as the dependent variable, and the severity group as the explanatory variable. This corresponds to the difference of means of clr-transformed taxa abundance among groups. We included a pseudo-count of half of the minimum nonzero value per sample prior to clr transformation. Statistical significance in ANCOM was set as a threshold of 0.1 of false discovery rate (FDR) [48]. ANCOM performs ratios between all taxa and reports a statistic W representing the number of ratios in which the null hypothesis could be rejected at the given statistical threshold. We concluded that a taxon had enough statistical evidence of changing between the groups if: (1) the number of subhypotheses rejected (W) was above 60% of the total number of ratios of that taxon, or (2) a taxon was found as a structural zero. We repeated the same procedure using pathway relative abundance (over 20% of prevalence and 0.1% of mean relative abundance). The same process was applied to correlate taxa and PPDdeepest/PPDmean.

Systemic inflammation markers, such as IL-6, IL-1Ra, WBC, and hsCRP, were correlated with bacterial taxa and pathways that were found to be significantly associated with PD. Correlations were performed independently in pocket and saliva samples. We used a linear model, including the participant’s age as a covariate, the inflammatory marker as the dependent variable, and the clr-transformed abundance metric as the covariate of interest.

### 4.9. Strain-Level Investigation

The genetic variability below the species level was examined using StrainPhlAn (as part of MetaPhlAn v2.7.2) [49]. StrainPhlAn is a marker-based approach to perform phylogenetic analysis within metagenomic species. We used the aligned reads to MetaPhlAn’s markers to reconstruct sample-specific markers that represent the most abundant strain per metagenomic species (sample2markers.py, accessed on 9 April 2021). Then, we used strainphlan.py to perform a multiple sequence alignment (MUSCLE v3.8.31) [50] and phylogenetic reconstruction (RaxML v8.1.5 [51,52], using GTRCAT substitution model) in 91 StrainPhlAn-detected clades. StrainPhlAn could successfully build 26 clades. We then created a phylogenetic distance matrix between samples in each of the available trees (ape and cophenetic function) and used it to build a PCoA (ape) and perform PERMANOVA (vegan). PERMANOVA tested whether disease severity was associated with the observed phylogenetic distances between participants within taxa. If samples for both blood and pocket sources were available, PERMANOVA included the tissue location as the covariate and participant ID as the strata.

## 5. Conclusions

Microbial diversity and species richness do not differ between severe PD and mild PD in either saliva or periodontal pockets. However, there are significant differences in the microbial composition between severe and mild PD in the subgingival microbiome (i.e., pocket samples) and, in a lower grade, in saliva, and this is positively associated with systemic inflammatory markers. Our data suggest that systemic inflammation in severe periodontitis may be driven by the oral microbiome and may support the indirect (inflammatory) mechanism for association between periodontitis and cardiovascular disease.

## Figures and Tables

**Figure 1 ijms-22-05876-f001:**
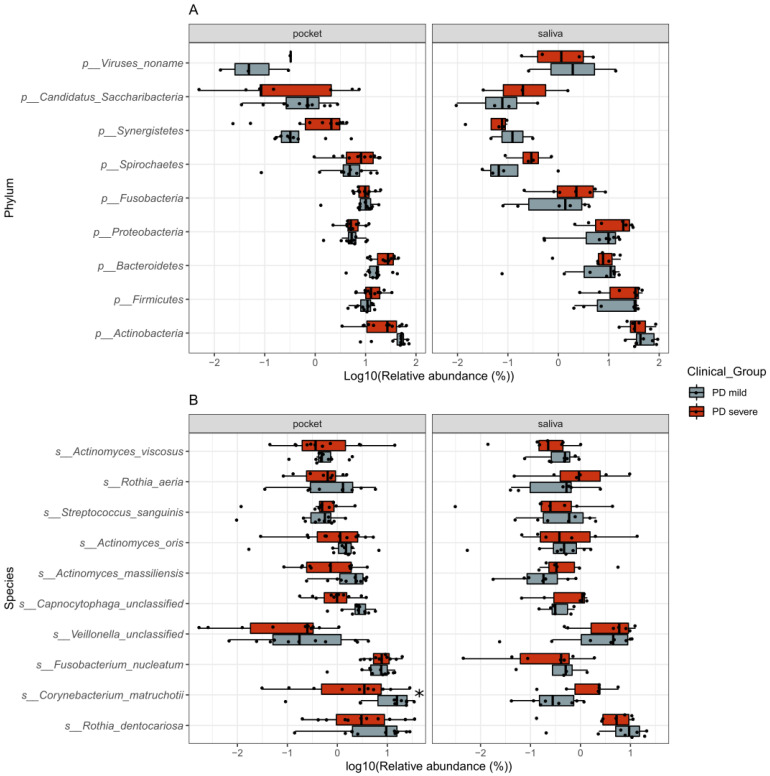
Taxa composition of pocket and saliva samples in each clinical group at the top 10 phylum (**A**) and species (**B**), in pocket (*n* = 22, 11 mild PD cases and 11 severe PD cases) and saliva (*n* = 15, 8 mild and 7 severe PD cases). Each dot represents the log10 of the relative abundance of taxa in a sample. The distribution of the abundances per clinical group is shown as a Tukey’s boxplot. Samples where the taxa are absent are not shown in the distribution. Significant differences are marked with an asterisk *.

**Figure 2 ijms-22-05876-f002:**
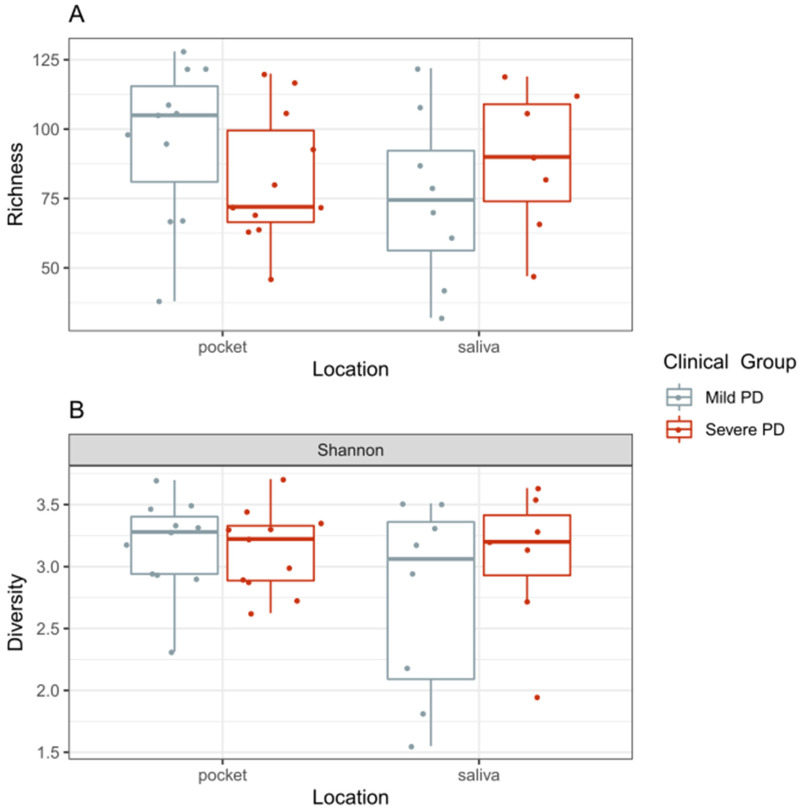
(**A**) Distribution of species richness in each of the studied groups (Kruskal–Wallis, *p* = 0.39). (**B**) Distribution of Shannon diversity in each group (Kruskal–Wallis, *p* = 0.81). Color represents the clinical group.

**Figure 3 ijms-22-05876-f003:**
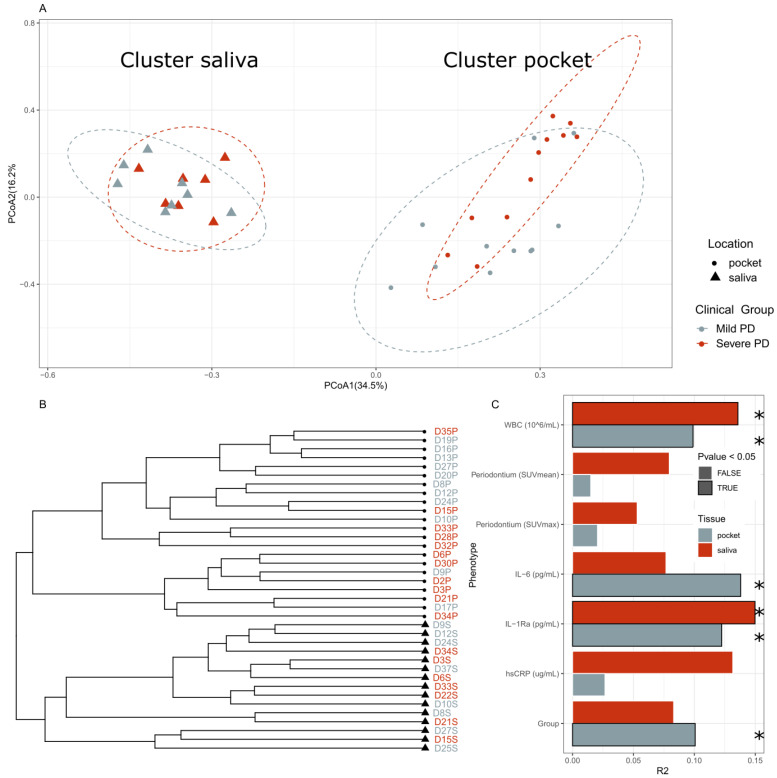
Phenotypic effect on the oral microbiome. (**A**) PCoA plot using the Bray–Curtis dissimilarity. The percentage of the axis represents the amount of variance explained by the principal component (PERMANOVA, saliva clinical group, *p* = 0.25, pocket clinical group, *p* = 0.03). (**B**) Dendrogram of hierarchical clustering based on Bray–Curtis dissimilarity. The shape represents the location of the sample, while the color represents whether the group was with mild or severe PD. (**C**) PERMANOVA-based variance explained from the microbiome bacterial species composition by different phenotypical parameters of the participants. R2 and P-values were calculated by PERMANOVA while controlling for the read number. An independent analysis was performed in pocket (grey) and saliva (red) samples. The factors found to have a *p*-value below 0.05 are outlined in black and marked with an asterisk *.

**Figure 4 ijms-22-05876-f004:**
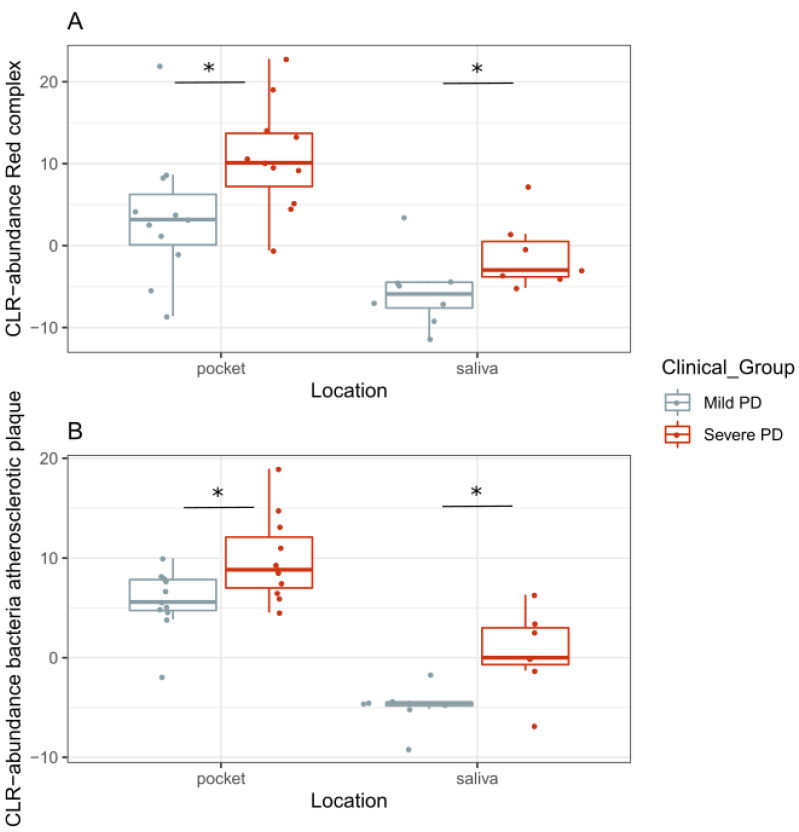
CLR-transformed relative abundances of (**A**) “red complex,” composed by *Porphyromonas gingivalis*, *Tannerellaforsythia*, and *Treponema denticola*, and of (**B**) bacteria found to be present in atherosclerosis plaque: *Propionibacterium propionicum*, *Parvimonas micra*, and *Parvimonas unclassified*. Clinical group is represented by different colors, where the x-axis shows the location where the sample was taken from. Significant differences are marked with an asterisk *.

**Table 1 ijms-22-05876-t001:** Baseline characteristics. Demographics are reported as mean ± SD or (number of participants). Circulating cytokines and chemokines are reported as median (IQR). One sample’sIL-6 concentration was under the detection limit of 0.70 pg/mL. Periodontal inflammation is reported as mean ± SD. Demographics were compared using the independent samples T-test (continuous normally distributed variables) and X^2^ test (categorical variables). Circulating cytokines, chemokines, and periodontal inflammation were compared using one-way ANCOVA while accounting for participant age. ^: *p* < 0.10, *: *p* < 0.05, **: *p* < 0.01. PPD: probing pocket depth, WBC: white blood cell count, SUV: standard uptake value of periodontal [^18^F] FDG-uptake on [^18^F]FDG PET/CT scan.

Demographics	Mild PD (*n* = 13)	Severe PD (*n* = 12)
DPSI, 0–4	**3.0 ± 0.0**	**4.0 ± 0.1 ****
Mean PPD, mm	**3.8 ± 0.5**	**5.3 ± 1.2 ****
Deepest PPD, mm	**4.7 ± 0.8**	**6.6 ± 1.6 ****
Age, years	**56 ± 9**	**63 ± 6 ***
Sex, % men (*n*)	62 (8)	25 (3)
BMI, kg/m^2^	25.1 ± 3.5	28.4 ± 7.0
Hypertension, % (*n*)	46 (6)	58 (7)
Current smoking, % (*n*)	15 (2)	8 (1)
**Circulating inflammatory markers**		
WBC, 10^6^/mL	5.8 [4.5–6.2]	5.7 [5.5–6.5]
IL-1Ra, pg/mL	**177 [157–236]**	**254 [188–428] ^**
IL-6, pg/mL	**1.4 [1.0–1.8]**	**2.0 [1.7–3.0] ^**
hsCRP, µg/mL	0.7 [0.3–1.7]	0.7 [0.6–1.2]
**Periodontal inflammation**		
Periodontium, SUVmean	**1.5 ± 0.2**	**1.7 ± 0.3 ^**
Periodontium, SUVmax	3.6 ± 0.5	3.6 ± 0.8

**Table 2 ijms-22-05876-t002:** Oral bacteria previously reported in the literature to be associated with PD and CVD. W values provided by ANCOM per bacterial community source (pocket/saliva) indicate the number of ratios that pass a significance threshold for each species (see Methods). The direction of the effect size, assessed by a clr-normalized linear model, is indicated, showing whether the species was more or less abundant in severe PD. The species filtered out due to low prevalence in the population are indicated. ‘Structural 0’ indicates the absence of the taxa in one disease group due to putative biological differences between the groups.

Taxa	Associated with	Source	ANCOM W Pocket	Direction Pocket	ANCOM W Saliva	Direction Saliva
*Aggregatibacter actinomycetemcomitans*	CVD-atherosclerotic plaques/aggressive PD	[13]/[14]	Low prevalence	-	Low prevalence	-
*Anaeroglobus geminatus*	CVD-atherosclerosis/PD	[13,14,15,16]	0	Increased	Structural	Increased
*Bulleidia extructa*	progressive PD	[17]	35	Increased	Structural	Increased
*Capnocytophaga gingivales*	CVD-blood lipid markers	[4]/[15]	0	Decreased	0	Increased
*Capnocytophaga granulosa*	CVD-blood lipid markers	[15]	0	Decreased	0	Increased
*Capnocytophaga ochracea*	Stable PD/CVD-blood lipid markers	[15]/[17]	0	Decreased	Structural	Increased
*Capnocytophaga unclassified*	CVD-blood lipid markers	[15]	112	Decreased	0	Increased
*Catonella morbi*	CVD-blood lipid markers/PD	[14,15]	2	Increased	Low prevalence	-
*Eubacterium brachy*	stable PD	[4,16,18]	105	Increased	0	Increased
*Eubacterium infirmum*	progressive PD	[17]	3	Increased	0	Increased
*Filifactor alocis*	progressive PD	[13,14,16,17]	0	Increased	0	Decreased
*Olsenella uli*	progressive PD	[17]	0	Increased	Structural	Increased
*Parvimonas micra*	CVD-uCRP/PD	[4]/[4,5,6,7,8,9,10,11,12,13,14,15,16]	4	Increased	Structural	Increased
*Porphyromonas gingivalis*	CVD-atherosclerotic plaques/PD	[13,14] /[16,19,20,21]	0	Increased	Structural	Increased
*Propionibacterium propionicum*	CVD-atherosclerotic plaques/progressive PD	[17]/[21]	0	Increased	0	Increased
*Tannerella forsythia*	CVD-atherosclerotic plaques/PD	[13,14] /[16,19]	0	Increased	0	Increased
*Treponema denticola*	PD	[14,16,19]	0	Increased	0	Increased
*Streptococcus*	CVD-HDL, ApoA1/PD	[21,22]	0	Increased	0	Decreased
*Neisseria*	CVD-negatively with HDL, ApoA1/PD	[21,22]	0	Increased	0	Increased
*Fusobacterium*	CVD-total cholesterol, LDL/PD	[14]/[16,21]	0	Increased	0	Increased

## Data Availability

Raw sequencing data were submitted to ENA with the study ID PRJEB42701. The complete processing pipeline and R scripts for analyses are available at https://github.com/GRONINGEN-MICROBIOME-CENTRE/Groningen-Microbiome/tree/master/Projects/Oral_microbiome_in_PD_2020 (accessed on 9 April 2021).

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
