# Peer review of "Oral Microbiome in Relation to Periodontitis Severity and Systemic Inflammation"

_ijms, 2021, doi:10.3390/ijms22115876_

Round 1

Reviewer 1 Report

The work by Plachokova et al. presents a new study of the oral microbiome from patients with periodontitis and the associations with cardiovascular disease. Although the question is of interest, the low sample size of the study and the missing control samples make it difficult to properly evaluate the results presented by the authors. In particular, both in the methods and in several paragraphs in the text, the authors refer to functional potential data profiled with HUMAnN, but only Table S3 contains the results which are not easy to interpret. Finally, from the discussion and conclusion paragraphs, it is not clear what the main finding/message of the work is, in particular related to cardiovascular disease.

Major

- The use of bar plot in Fig. 1 is a bit misleading, especially considering that I believe many relative abundance values are very close to zero. I would suggest the use of boxplot to better represent the distributions of relative abundances of different species and maybe the use of log-scale for the relative abundance values would help in visualizing when differences are with values close to zero

- while the data is from shotgun metagenomics no results trying to investigate below the species level diversity are present. It would be interesting to check whether the differential species between mild and severe PD are very similar or completely different strains within subjects of the same group. This would support a microbial signature for periodontitis.

- lines 118/120, it is stated that the number of permutations for the PERMANOVA analysis was set to 2000 (instead of the default 999) and this seems quite a high number. Was it set to enforce having low-p-values?

- lines 237/238 in the discussion argued that shotgun metagenomics has the advantage to study the functional potential of a microbiome sample, but I could not find this discussed enough in the results, and in particular, within that paragraph, the sentence doesn't really fit as the main point to the paragraph is actually discussing cardiovascular markers. 

Minor

- typo Abstract line 14 "The aim is of this work"

- line 85 "General description" I believe should be removed

- lines 103/142/144/145/170/171/175/176 species names and genera should be italicized

- the above list is now exhaustive, reports only the instances I was able to identify

Reviewer 2 Report

The manuscript is focusing on the relationship between periodontitis and systemic inflammation using the oral microbiome based on metagenomic shotgun-sequencing and circulating inflammatory markers.

Authors claimed that there were remarkable differences severe and mild periodontitis in the oral microbiome from subgingival pocket and saliva samples and the microbiome profiles were positively associated with systematic inflammatory markers. In overall, the idea for the analysis and the findings in the present study are interesting, but I want to comment out several points for the manuscript.

  1. In the introduction, author stated that “To date, there are limited studies focused on the oral microbiome in PD that use metagenomic shotgun-sequencing (MGS)” and only two references were provided. But there are many studies about oral microbiome in the periodontitis using metagenomics technologies (More than 30 studies). Please correct the statement and refer a proper review paper for the microbiome studies.

  1. In table 1, which statistical tests were used to compare two groups, mild PD and severe PD.

  1. Although authors stated the small sample size as the limitation of the current study in the discussion section, please provide the justification of the study sample described in the Material and methods section.

  1. Which 16S rRNA database was used for the taxonomic assignments? The number of assigned species seems to be small. Please specify the database for the assignment. Also present the number of total sequencing reads per sample and the number of assigned OTUs (or features) per sample.

  1. How bacterial pathways were assigned in the microbiome analyses. Please provide detail processes such as assignment methods and metabolic pathway databases.

  1. In figure 2a, the authors showed the distribution of species richness. How about the statistical test for that? Which test was used and was there no statistical significance? Please state the results in the main text.

  1. In line 117 ~ 120, authors described two different statistical tests to compare the mild and severe PD groups. It is clear that the former test was PERMANOVA. How about the latter one? What is “the independent analysis (2,000 permutations, P = 0.03)?

  1. The last result section, “2.2.5. PD bacteria and bacterial metabolic pathways associated with systemic inflammation”, author just described and stated the results in the main text. It should be great to show and summarize the results in this section as tables.

  1. In the main text and figures, the nomenclature for the bacteria is not appropriate. For example, (1) in line 103, “Corynebacterium matruchotii” should be italic, (2) similarly, the bacterial species names in the figure 1b should be italic. Please check and correct the mis-nomenclature for the whole manuscript.

  1. In line 22, Figure 4 => Figure 3

  1. Please provide p-values in the figure 3a.

Reviewer 3 Report

Please write in small letters titles of the references 

  1. 28, 29. 33, 36. 37.38 , 42,43.44, in  reference list. 

Round 2

Reviewer 1 Report

I thank the authors for revising their manuscript which I think improved clarity and readability.

Author Response

We appreciate very much the ciritical remarks of the reviewer that helped to improve the quality of our manuscript. Thank you for your time!